# Multi-task Learning with Iterative Training in Hybrid Labeling Dataset for Semi-supervised Abdominal Multi-organ and Tumor Segmentation

Zhiqiang Zhong[1,†][0000−0002−0220−159X], Rongxuan He[1,2,†][0009−0005−0407−6963], Deming Zhu[1][0000−0002−6747−0766], Mengqiu Tian[1][0009−0006−1355−6691], and Songfeng Li[1][0000−0002−5228−9630]⋆

[1] Percept Vis Med Technol Co LTD, Guangzhou 510275, People's Republic of China.
lisongfeng@pvmedtech.com
[2] Johns Hopkins University, Baltimore MD 21218, USA

**Abstract.** Simultaneous segmentation of organs and tumors from abdominal CT images is challenging, and the task has many critical clinical applications such as disease diagnosis, lesion and organ measurements, and surgical planning. Based on nnU-Net, we develop a method for abdominal organ and whole-body pan-tumor segmentation for both abdominal and whole-body CT images. First, in a fully supervised setting, we train the base models of organs and tumors to generate initial pseudo-labels. Then, in a semi-supervised setting, a mixed-labeled dataset is used to iteratively train a higher-performance segmentation model to create higher-quality pseudo-labels. Due to the correlation between organs and tumors in the abdominal region, we leverage the idea of multi-task learning to train a single model to segment both organs and tumors to improve the performance of a single task. Finally, to trade off segmentation efficiency and accuracy, we design a sliding window strategy based on the body prior and a simplified version of test-time augmentation (TTA4). Our final model achieved 88.93% mean organ DSC and 45.76% tumor DSC on the FLARE23 online validation set. In addition, the average running time and area under GPU memory-time curve were 26.7s and 49352.9MB, respectively. On the test set, we achieved mean organ and tumor DSC of 89.68% and 62.89%, respectively, NSD of 95.89% and 51.69%, respectively, and average inference time of 18.53s. Our code is publicly available at https://github.com/LeoZhong997/FLARE23.

**Keywords:** Segmentation · Multi-task learning · Semi-supervised learning.

## 1 Introduction

Simultaneous segmentation of organs and tumors from abdominal CT images is a formidable challenge that holds immense clinical significance. It plays a pivotal role in various critical clinical applications, such as disease diagnosis, precise

---

⋆ Corresponding author

lesion and organ measurements, and the development of surgical plans. Nevertheless, manually labeling organs and lesion locations is a time-consuming task that demands a great deal of expertise from physicians. FLARE23 is a challenge aimed at fostering the development of fully automatic solutions for this task. Expanding upon the 13 abdominal organs segmentation task of FLARE22 [13], FLARE23 requires participants to simultaneously segment tumors, a more practical study given that the majority of real clinical data may contain lesions. Furthermore, the challenge restricts the inference time and GPU memory usage to mimic actual clinical conditions, implying that we cannot complete the task solely by increasing the model size or using more computational resources.

Semi-supervised learning is a crucial strategy employed in medical image segmentation tasks, due to the limited availability of medical data and the time-consuming annotation process. One of the most common approaches to semi-supervised segmentation is to use pseudo-labels [9] generated by a model trained on the labeled data. When training a model with a large amount of unlabeled data, the accuracy of the pseudo-labels becomes critical. Consequently, eliminating uncertain pseudo-labels is a vital step in the training procedure. The standard method for filtering out uncertain pseudo-labels involves applying a confidence threshold to determine whether the pseudo-labels are reliable. Furthermore, recent studies have demonstrated that these unreliable pseudo-labels can also be leveraged in the self-training process [17].

In this paper, we propose an iterative training framework based on nnU-net to perform organ and tumor segmentation tasks. We start from a single-task setting, where we iteratively train the organ segmentation model. Semi-supervised learning is employed to generate pseudo labels for the partially labeled data and unlabeled data. Subsequently, we transition to a multi-task setting, training a model to perform both organ and tumor segmentation tasks using the pseudo labels generated in the prior stage. Additionally, we incorporate unlabeled data into the training set. Furthermore, to enhance inference speed, we introduce a sliding window strategy and we utilize a simplified version of test-time augmentation (TTA4) to improve segmentation accuracy.

## 2   Method

### 2.1   Preprocessing

The preprocessing strategies we use are as follows:

- Data cleaning or statistical analysis:
  We perform label analysis to check label completeness. Out of 2200 labeled data, 222 cases include complete organ labels without tumors, and 1497 cases have tumor labels. These two subsets are utilized for training our single-task models.
- Reorientation:
  As we want the network to predict images regardless of orientation, we reorient the images to the standard RAS orientation during the training phase.

Later, we will apply mirroring operations in the later stages of data augmentation to enhance the network's orientation robustness.
- Resampling method for anisotropic data:
  In order to leverage the physical information within the CT data, all images are resampled to the same resolution of 4.0mm × 1.2mm × 1.2mm.
- Intensity normalization method:
  Initially, we compute the 0.5 and 99.5 percentiles, as well as the mean and standard deviation of the data intensity. Subsequently, the data is clipped to the 0.5 and 99.5 percentiles, and z-score normalization is applied using the global mean and standard deviation.

## 2.2 Proposed Method

We introduce an iterative training framework for the task of multi-organ and tumor segmentation. Our networks are derived from the 3D nnU-Net [8]. However, we separate from the nnU-Net's auto-configuration and introduce two fixed network architectures: the medium and large nnU-Net, with their parameters detailed in the experiment part. Fig. 1 illustrates the workflow of our proposed approach. Our approach comprises two stages: single-task training and multi-task training.

**Single-task Training** During the single-task stage, we train the nnU-Net separately for organ and tumor segmentation. To address the multi-organ segmentation task, we utilize the 222 labeled data that include complete organ labels.

Following the development of the organ segmentation model, we employ it to generate pseudo labels for the remaining 1978 labeled data lacking organ labels. Nevertheless, within these 1978 labeled data, we have part of ground truth labels. We propose combining these ground truth labels with the pseudo labels. Since this model only performs organ segmentation, we filter out organs that do not contain tumors in the true labels. Determining the organ to which the tumor belongs is accomplished through morphological analysis. We conduct a morphological dilation operation on the tumors and if an overlap exists between the tumor and an organ, the tumor is attributed to that organ. Subsequently, we replace the corresponding pseudo labels with the ground truth labels for organs without tumors, resulting in a hybrid labeled dataset.

The hybrid labeled dataset is employed for training the organ segmentation model, and we utilize the model to generate pseudo labels for the entire 2200 training set. Iterative training is then conducted to enhance the accuracy of our pseudo labels of organs.

In the context of the tumor segmentation task, we utilize the 1497 labeled data containing tumor labels. However, due to suboptimal Dice Similarity Coefficient (DSC) and Normalized Surface Dice (NSD) performance, we do not employ this model in our subsequent training procedures.

**Multi-task Training** To reduce inference time costs and maximize the utilization of the correlation between organs and tumors, we suggest training a single model capable of accomplishing both organ and tumor segmentation tasks. The organ model trained in the previous stage is utilized to generate the pseudo labels for the 1497 labeled data. These pseudo labels are then combined with the ground truth, following the same procedure described earlier. Following the utilization of the hybrid labeled subset for training the multi-task model, we employ the model to generate the pseudo labels of the 2200 training set and retrain the model.

Once the multi-task model is trained using the 2200 labeled data, we employ the model to generate the pseudo labels of the 1800 unlabeled data. Subsequently, we straightforwardly add these data to the training set and conduct iterative training twice to obtain the final model.

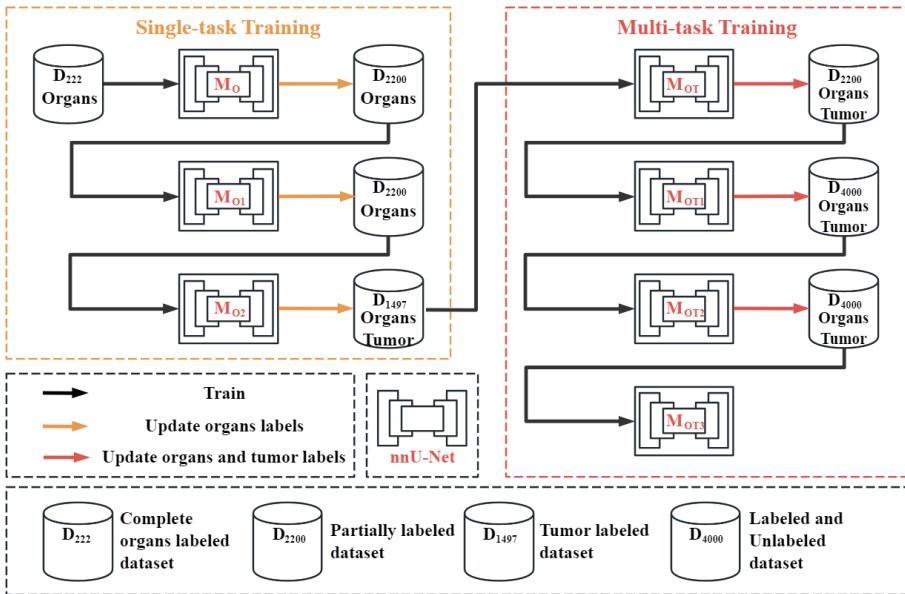

**Fig. 1.** Workflow of our proposed approach. The workflow comprises two stages: single-task training and multi-task training.

**Loss Function** We use the summation between a weighted Dice loss and cross-entropy loss because compound loss functions have been proven to be robust in various medical image segmentation tasks [10]. What's more, deep supervision is used to fully utilize the feature information of the intermediate encoding and decoding layers.

**Sliding Window Strategy** In order to improve inference speed and reduce resource consumption, we adopt the sliding window strategy to fuse the predictions of overlapping patches. We adapt the fast sliding window strategy initially proposed by the FLARE22 winning team [7] to align it with the requirements of the tumor segmentation task. Given that tumors can appear in various regions in the abdominal area, the absence of a label in the central patch does not necessarily imply the absence of tumors in the surrounding patches. Consequently, for every slice along the z-axis, after the acquisition of the central patch, we also retrieve all the surrounding patches to generate the final prediction.

### 2.3   Post-processing

To improve the performance of pseudo-labels, we employ connected component analysis on organs, retaining the largest 3D connected component. If the organ's Dice loss increases following connected component analysis, we opt to conduct the analysis for that specific organ. During the validation and testing phases, connected component analysis is deactivated to reduce time overhead.

Additionally, we introduce a streamlined test-time augmentation approach (TTA4). Instead of applying augmentation in all 8 directions, we restrict it to 4 directions: the original orientation and the flipped orientations along the x, y, and z axes, respectively.

## 3   Experiments

### 3.1   Dataset and evaluation measures

The FLARE 2023 challenge is an extension of the FLARE 2021-2022 [12][13], aiming to promote the development of foundation models in abdominal disease analysis. The segmentation targets cover 13 organs and various abdominal lesions. The training dataset is curated from more than 30 medical centers under the license permission, including TCIA [2], LiTS [1], MSD [16], KiTS [5][6], autoPET [4,3], TotalSegmentator [18], and AbdomenCT-1K [14]. The training set includes 4000 abdomen CT scans where 2200 CT scans with partial labels and 1800 CT scans without labels. The validation and testing sets include 100 and 400 CT scans, respectively, which cover various abdominal cancer types, such as liver cancer, kidney cancer, pancreas cancer, colon cancer, gastric cancer, and so on. The organ annotation process used ITK-SNAP [19], nnU-Net [8], and MedSAM [11].

The evaluation metrics encompass two accuracy measures—Dice Similarity Coefficient (DSC) and Normalized Surface Dice (NSD)—alongside two efficiency measures—running time and area under the GPU memory-time curve. These metrics collectively contribute to the ranking computation. Furthermore, the running time and GPU memory consumption are considered within tolerances of 15 seconds and 4 GB, respectively.

## 3.2   Implementation details

**Environment settings**  The development environments and requirements are presented in Table 1. The training protocols of medium nnU-Net and large nnU-Net are listed in Table 2 and Table 3 respectively. We adopt data augmentation of rotation, scaling, Gaussian noise and blur, brightness, contrast, gamma, elastic deformation, and mirror on the fly during training. Notably, we reduced the number of test time augmentation(TTA) flips to balance segmentation accuracy and inference time.

**Table 1.** Development environments and requirements.

| | |
|---|---|
| System | Ubuntu 18.04.5 LTS |
| CPU | Intel(R) Xeon(R) Gold 6326 CPU @ 2.90GHz |
| RAM | 8×64GB; 3200MT/s |
| GPU (number and type) | Four NVIDIA A4000 16G |
| CUDA version | 11.6 |
| Programming language | Python 3.8.13 |
| Deep learning framework | torch 1.13, torchvision 0.14.0 |
| Specific dependencies | nnU-Net 1.7.0 |
| Code | https://github.com/LeoZhong997/FLARE23 |

**Table 2.** Training protocols for medium nnU-Net

| | |
|---|---|
| Network initialization | "He" normal initialization |
| Batch size | 2 |
| Stage number | 5 |
| Convolution number per stage | 2 |
| Patch size | 32×128×192 |
| Total epochs | 1500 |
| Optimizer | SGD with nesterov momentum (μ = 0.99) |
| Initial learning rate (lr) | 0.01 |
| Lr decay schedule | Poly learning rate policy: $(1 - epoch/1000)^{0.9}$ |
| Training time | 25 hours |
| Loss function | Dice loss and cross-entropy loss |
| Number of model parameters | 22M |
| Number of flops | 253.90G |
| $CO_2$eq | 8.14 Kg |

**Table 3.** Training protocols for large nnU-Net

| | |
|---|---|
| Network initialization | "He" normal initialization |
| Batch size | 2 |
| Stage number | 6 |
| Convolution number per stage | 3 |
| Patch size | $32\times128\times192$ |
| Total epochs | 1500 |
| Optimizer | SGD with nesterov momentum ($\mu = 0.99$) |
| Initial learning rate (lr) | 0.01 |
| Lr decay schedule | Poly learning rate policy: $(1 - epoch/1000)^{0.9}$ |
| Training time | 33 hours |
| Loss function | Dice loss and cross-entropy loss |
| Number of model parameters | 85M |
| Number of flops | 375.14G |
| $CO_2$eq | 9.83 Kg |

## 4  Results and discussion

### 4.1  Quantitative results on validation set

First, we train the single models $M_O$ and $M_T$ on the fully-labeled dataset of 222 cases of organ and 1497 cases of tumor, respectively. To obtain complete labels of organs, $M_O$ first generates pseudo-labels on partially-labeled data of 2200 cases and combines them with ground true labels to produce a mixed-labeled organ dataset for training $M_{O1}$, and continues to iterate to generate a new dataset for training $M_{O2}$, to produce high-quality organ pseudo labels.

To validate the effectiveness of multi-task segmentation, we combine the mixed labels of organs with the ground true label of tumor on 1497 cases to train the model $M_{OT}$, which is able to segment all organs and tumor at once and achieves better segmentation performance than single-task segmentation.

Further, $M_{OT}$ was utilized to generate new organs and tumor pseudo-labels on 2200 images and combined with ground true labels to form a hybrid-label dataset, wherein, due to the low accuracy of tumor segmentation, we utilized organs to constrain tumor pseudo-labels during label merging, and disregarded the results of tumor segmentation outside of organs. Using this dataset, we trained the model $M_{OT1}$.

In order to verify the effectiveness of unlabeled data on model segmentation performance improvement, we use $M_{OT1}$ to generate segmentation results on 4000 cases, of which 2200 cases are regenerated as a mixed-labeled dataset on partially-labeled data. The remaining 1800 cases are directly used as pseudo-labels for unlabeled data. We train the model $M_{OT2}$ on these 4000-cases dataset.

Finally, we utilize $M_{OT2}$ to iterate on the 4000-cases to generate a new dataset and upgrade the medium model to large to extract more feature information, then train the final model $M_{OT3}$. In order to balance the inference speed and

segmentation accuracy, we adopt the TTA4 strategy (by reducing the number of flips of TTA, i.e., flipping the input image over x, y, and z, respectively) to complete the final inference process.

We report the final results of DSC and NSD of organ and tumor on the validation set in Table 4. The results of ablation studies to analyze the effect of multi-task segmentation and unlabeled data can be obtained from Table 5.

**Table 4.** Quantitative evaluation results. The public validation denotes the performance on the 50 validation cases with ground truth. Please present both the mean score and standard deviation. The online validation denotes the leaderboard results. The Testing results will be released during MICCAI.

| Target | Public Validation | | Online Validation | | Testing | |
|---|---|---|---|---|---|---|
| | DSC(%) | NSD(%) | DSC(%) | NSD(%) | DSC(%) | NSD (%) |
| Liver | 97.46±0.48 | 99.26±1.32 | 97.44 | 99.15 | 96.72 | 98.33 |
| Right kidney | 94.35±8.27 | 97.4±7.01 | 93.56 | 95.61 | 94.05 | 95.16 |
| Spleen | 96.51±0.69 | 99.76±0.58 | 96.72 | 99.23 | 96.06 | 98.43 |
| Pancreas | 86.14±5.42 | 98.38±2.79 | 85.49 | 96.99 | 90.05 | 98.4 |
| Aorta | 94.68±1.67 | 98.09±2.37 | 95.38 | 98.94 | 95.58 | 99.43 |
| Interior vena cava | 92.94±1.66 | 96.69±2.69 | 93.94 | 97.27 | 94.51 | 98.37 |
| Right adrenal gland | 80.21±12.36 | 96.66±14.01 | 79.74 | 93.8 | 79.15 | 93.58 |
| Left adrenal gland | 80.48±5.9 | 97.54±2.89 | 79.62 | 93.67 | 79.16 | 93.66 |
| Gallbladder | 81.95±24.98 | 88.32±27.02 | 80.87 | 81.42 | 79.89 | 82.11 |
| Esophagus | 81.09±14.86 | 94.51±14.65 | 82.41 | 94.39 | 87.46 | 98.45 |
| Stomach | 92.69±3 | 98.37±3.27 | 93.37 | 98.45 | 93.07 | 98.41 |
| Duodenum | 83.46±6.07 | 96.49±4.48 | 84.38 | 96.33 | 88.09 | 98.10 |
| Left kidney | 93.99±6.21 | 96.33±8.33 | 93.22 | 95.24 | 92.90 | 94.62 |
| Tumor | 52.23±35.08 | 51.8±34.38 | 45.76 | 38.5 | 62.89 | 51.69 |
| Average | 86.30 | 93.54 | 85.85 | 91.36 | 87.83 | 92.77 |

### 4.2   Qualitative results on validation set

Fig. 2 shows four representative segmentation results of the final model $M_{OT3}$ in the validation dataset. For Case #FLARETs_0083 and Case #FLARETs_0027, the model successfully identified all organs and accurately segmented the tumor boundaries. For Case #FLARETs_0051, although the model had identified all the correct organs, it failed to successfully segment the tumor, resulting in lower metrics for both the tumor and the organs. In Case #FLARETs_0091, the model even failed to determine the location of the prostate tumor. We believe that, on the one hand, there is no annotation information for prostate organs in the dataset, resulting in the failure to establish a connection between organ and tumor; on the other hand, prostate tumors are a low percentage in the dataset, and the model lacks sufficient data to learn to segment this target.

**Table 5.** DSC(%) and NSD(%) of organs and tumors on online validation set.

| Model | Training images | Metrics | Liver | RK | Spleen | Pancreas | Aorta | IVC | RAG | LAG | Gallbladder | Esophagus | Stomach | Duodenum | LK | Tumor | organ Mean |
|---|---|---|---|---|---|---|---|---|---|---|---|---|---|---|---|---|---|
| $M_O$ | 222 | DSC | 95.72 | 90.75 | 94.14 | 82.41 | 94.79 | 92.65 | 78.73 | 77.09 | 78.00 | 80.14 | 91.00 | 81.55 | 89.88 | - | 86.68 |
| | | NSD | 97.48 | 92.72 | 95.09 | 95.71 | 97.96 | 95.55 | 93.53 | 92.03 | 76.50 | 92.72 | 95.19 | 94.34 | 91.95 | - | 93.14 |
| $M_{O1}$ | 2200 | DSC | 96.65 | 91.80 | 95.97 | 83.49 | 95.19 | 93.57 | 79.50 | 78.08 | 82.24 | 81.85 | 92.30 | 82.74 | 89.01 | - | 87.88 |
| | | NSD | 98.14 | 93.91 | 98.03 | 96.13 | 98.73 | 96.75 | 93.67 | 92.34 | 81.92 | 93.97 | 96.91 | 95.20 | 91.23 | - | 94.38 |
| $M_{O2}$ | 2200 | DSC | 97.08 | 92.03 | 96.63 | 84.90 | 95.43 | 94.06 | 79.97 | 79.53 | 81.43 | 82.57 | 92.79 | 84.12 | 88.41 | - | 88.38 |
| | | NSD | 98.67 | 94.61 | 99.04 | 96.83 | 99.02 | 97.44 | 94.13 | 93.80 | 81.56 | 94.73 | 97.76 | 96.13 | 91.85 | - | 95.04 |
| $M_T$ | 1497 | DSC | - | - | - | - | - | - | - | - | - | - | - | - | - | 34.34 | - |
| | | NSD | - | - | - | - | - | - | - | - | - | - | - | - | - | 24.02 | - |
| $M_{OT}$ | 1497 | DSC | 97.32 | 92.62 | 96.46 | 84.56 | 95.19 | 93.61 | 79.84 | 79.46 | 81.18 | 81.88 | 93.12 | 83.50 | 92.88 | 43.76 | 88.59 |
| | | NSD | 99.00 | 94.68 | 98.85 | 96.62 | 98.64 | 96.72 | 93.89 | 93.62 | 81.38 | 93.94 | 98.13 | 95.86 | 94.86 | 36.22 | 95.09 |
| $M_{OT1}$ | 2200 | DSC | 97.42 | 93.44 | 96.64 | 85.23 | 95.29 | 93.85 | 79.63 | 79.34 | 81.45 | 82.26 | 93.05 | 83.98 | 93.47 | 44.31 | 88.85 |
| | | NSD | 99.17 | 95.48 | 99.09 | 96.88 | 98.80 | 97.21 | 93.95 | 93.63 | 81.82 | 94.52 | 98.03 | 96.13 | 95.41 | 37.49 | 95.39 |
| $M_{OT2}$ | 4000 | DSC | 97.43 | 93.67 | 96.63 | 84.99 | 95.28 | 93.88 | 80.77 | 79.60 | 80.61 | 82.12 | 93.08 | 83.83 | 93.32 | 44.73 | 88.86 |
| | | NSD | 99.22 | 95.84 | 99.02 | 96.77 | 98.79 | 97.24 | 94.79 | 93.64 | 80.92 | 94.18 | 98.09 | 96.03 | 95.27 | 37.41 | 95.37 |
| $M_{OT3}$ | 4000 | DSC | 97.44 | 93.56 | 96.72 | 85.49 | 95.38 | 93.94 | 79.74 | 79.62 | 80.87 | 82.41 | 93.37 | 84.38 | 93.22 | 45.76 | 88.93 |
| | | NSD | 99.15 | 95.61 | 99.23 | 96.99 | 98.94 | 97.27 | 93.80 | 93.67 | 81.42 | 94.39 | 98.45 | 96.33 | 95.24 | 38.50 | 95.42 |

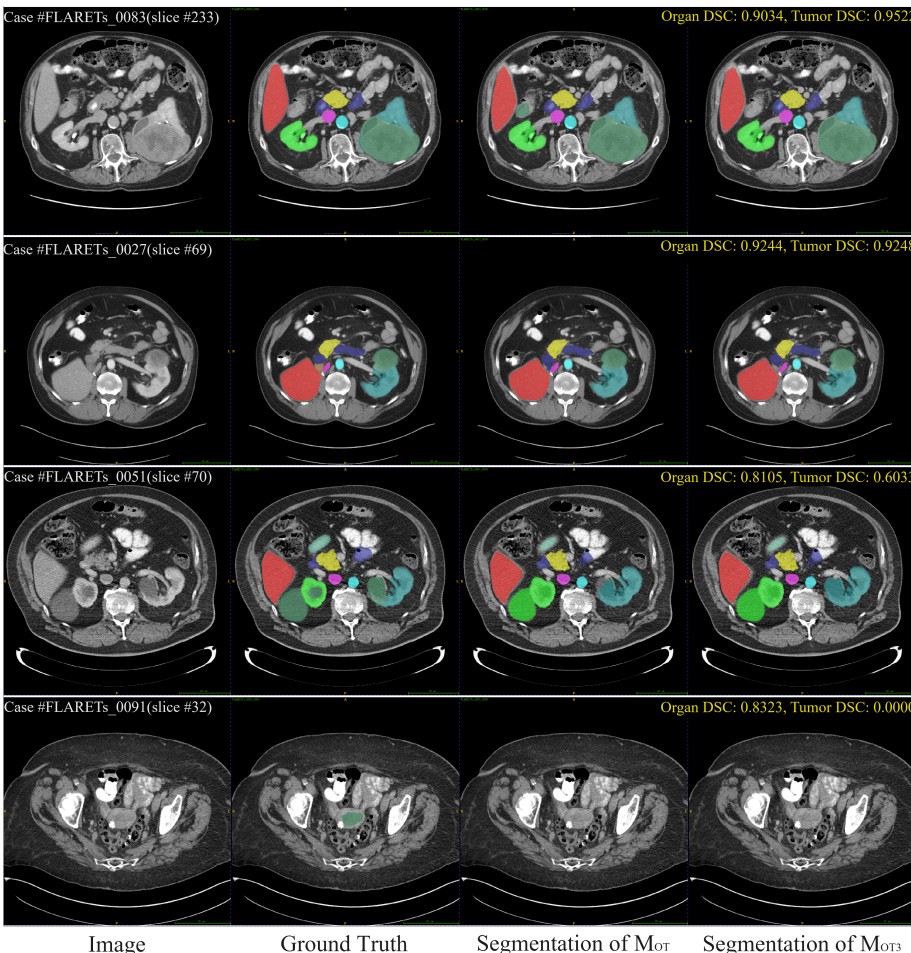

**Fig. 2.** Qualitative results of our final model on two easy cases and two hard cases.

### 4.3 Segmentation efficiency results on validation set

We applied a sliding window strategy with body prior and a simplified TTA4 method on the final model $M_{OT3}$ to build the final submitted docker image. In Table 6 and Table 7, we report the efficiency evaluation results from the official platform.

**Table 6.** Quantitative evaluation of segmentation efficiency in terms of the running time and GPU memory consumption. Total GPU denotes the area under GPU Memory-Time curve. Evaluation GPU platform: NVIDIA QUADRO RTX5000 (16G).

| Case ID | Image Size | Running Time (s) | Max GPU (MB) | Total GPU (MB) |
|---------|------------|------------------|--------------|----------------|
| 0001 | (512, 512, 55) | 19.28 | 2468 | 31332 |
| 0051 | (512, 512, 100) | 26.05 | 2468 | 49968 |
| 0017 | (512, 512, 150) | 38.61 | 2468 | 54208 |
| 0019 | (512, 512, 215) | 23.93 | 2468 | 43972 |
| 0099 | (512, 512, 334) | 27.27 | 2468 | 51457 |
| 0063 | (512, 512, 448) | 31.75 | 2468 | 59780 |
| 0048 | (512, 512, 499) | 34.23 | 2468 | 65627 |
| 0029 | (512, 512, 554) | 38.02 | 2468 | 73601 |

**Table 7.** Efficiency evaluation results of our submitted docker. All metrics reported are the average values on 20 validation cases.

| Time | GPU Memory | AUC GPU Time | CPU Utilization | AUC CPU Time | RAM | AUC RAM Time |
|------|-----------|--------------|-----------------|--------------|-----|--------------|
| 26.7 | 2504.6 | 49352.9 | 66.67 | 916.63 | 6283.97 | 126713.2 |

### 4.4 Results on final testing set

Our method achieved seventh place out of 37 submissions in the final testing set. Tables 4 and 8 show the detailed evaluation metrics of our method in the final testing set.

**Table 8.** Testing results of our proposed method. All metrics reported are the average values on 400 testing cases.

| Organ DSC | Organ NSD | Tumor DSC | Tumor NSD | AUC GPU Time | Time |
|-----------|-----------|-----------|-----------|--------------|------|
| 0.8968 | 0.9589 | 0.6289 | 0.5169 | 33804 | 18.53 |

## 4.5   Limitation and future work

We used a simple but effective iterative training strategy to gradually improve the quality of pseudo-label generation, but there may be noise in the pseudo-labels, which can limit or even degrade the segmentation performance of the model. Therefore, we will investigate the latest pseudo-label selection strategy in our future work to form a positive feedback loop in iterative training.

## 5   Conclusion

In this paper, we iteratively train a model capable of segmenting both abdominal organs and whole-body pan-tumors on a mixed-labeled dataset based on the nnU-Net framework, which combines fully supervised, semi-supervised, and multi-task learning. In addition, this paper designs a sliding window strategy based on the body prior and a simplified test-time augmentation to trade-off efficiency and accuracy during inference. The results of the public validation set of FLARE2023 show that the method has good segmentation performance and computational efficiency.

**Acknowledgements** The authors of this paper declare that the segmentation method they implemented for participation in the FLARE 2023 challenge has not used any pre-trained models nor additional datasets other than those provided by the organizers. The proposed solution is fully automatic without any manual intervention. We thank all the data owners for making the CT scans publicly available and CodaLab [15] for hosting the challenge platform.

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

**Table 9.** Checklist Table. Please fill out this checklist table in the answer column.

| Requirements | Answer |
| --- | --- |
| A meaningful title | Yes |
| The number of authors ($\leq$6) | 5 |
| Author affiliations, Email, and ORCID | Yes |
| Corresponding author is marked | Yes |
| Validation scores are presented in the abstract | Yes |
| Introduction includes at least three parts: background, related work, and motivation | Yes |
| A pipeline/network figure is provided | Fig. 1 |
| Pre-processing | Page 2 |
| Strategies to use the partial label | Page 3 |
| Strategies to use the unlabeled images. | Page 4 |
| Strategies to improve model inference | Page 5 |
| Post-processing | Page 5 |
| Dataset and evaluation metric section is presented | Page 5 |
| Environment setting table is provided | Table 1 |
| Training protocol table is provided | Table 2 and Table 3 |
| Ablation study | Page 7 |
| Efficiency evaluation results are provided | Table 6 and Table 7 |
| Visualized segmentation example is provided | Fig. 2 |
| Limitation and future work are presented | Yes |
| Reference format is consistent. | Yes |