# OpenReview forum: "Multi-task Learning with Iterative Training in Hybrid Labeling Dataset for Semi-supervised Abdominal Multi-organ and Tumor Segmentation"
_MICCAI.org/2023/FLARE — Submitted to FLARE 2023_

### Official Review · Reviewer_xQmq · 2023-09-27
**review for 'Multi-task Learning with Iterative Training in Hybrid Labeling Dataset for Semi-supervised Abdominal Multi-organ and Tumor Segmentation'**

**Rating:** 7
**Confidence:** 5

**Review:**

Pros:　 The manuscript is generally well-written and easy to follow. The manuscript presents a novel training approach, according to the dataset distribution.
Cons:  There are some small typos.
1.	The abbreviation ‘OARS’ of Figure 2 should be described.
2.	The arrow at the bottom of ‘Organ segmentation’ needs to be redesigned in Figure 1.
3.	What does ‘4’ in ‘test-time augmentation approach (TTA4)’ mean?

---

> ### Author Response · Authors · 2023-11-06
> **Update Figures 1 and 2**
>
> 1. We have updated Figure 1 for a better understanding.
> 2. ‘OARS’  has changed to 'Organ' in Figure 2.
> 3. TTA4 is a simplified version of test-time augmentation(TTA). Instead of applying augmentation in all 8 directions, we restrict it to 4 directions: the original orientation and the flipped orientations along the x, y, and z axes, respectively.

---

### Official Review · Reviewer_Bm3n · 2023-09-27
**Review for 'Multi-task Learning with Iterative Training in Hybrid Labeling Dataset for Semi-supervised Abdominal Multi-organ and Tumor Segmentation'**

**Rating:** 7
**Confidence:** 5

**Review:**

This manuscript presents a two tasks method with five training steps to improve the efficiency and accuracy. The algorithm consists of two parts: organ segmentation and multi-task segmentation. However, I would like to ask the authors to address the following questions/remarks:

1.The grammar throughout the text needs to pay attention to verb tenses and singular and plural forms. e.g., ‘organ’ in 3rd line , 1st paragraph, section 4.1

2.For a better understanding, please ensure that the arrangement of arrows in the workflow (figure 1) is more logically coherent. This will help readers to more easily grasp the relationships and flow between different parts, thereby enhancing the clarity and readability of the article.

---

> ### Author Response · Authors · 2023-11-06
> **Update Figure 1 and Correcting grammar**
>
> 1. We have updated Figure 1 for a better understanding.
> 2. Check grammar throughout the text and correct it.

---

### Official Review · Reviewer_B2eQ · 2023-10-04
**Review for “Multi-task Learning with Iterative Training in Hybrid Labeling Dataset for Semi-supervised Abdominal Multi-organ and Tumor Segmentation”**

**Rating:** 9
**Confidence:** 4

**Review:**

The paper is a nice work with complete structure.  There are several minor issues:
1. The name of the legend should be attached below the Qualitative results in Figure 2.
2. It is suggested that the experimental results of ablation studies are shown in Figure 2 for comparison.

---

> ### Author Response · Authors · 2023-11-06
> **Update Figure 2**
>
> We have added the legend name and $\rm{M}_{\rm{OT}}$ results of the ablation study to Figure 2.

---

### Official Review · Reviewer_pJUa · 2023-10-04
**Multi-task Learning with Iterative Training in Hybrid Labeling Dataset for Semi-supervised Abdominal Multi-organ and Tumor Segmentation**

**Rating:** 8
**Confidence:** 4

**Review:**

The authors present a method for simultaneous segmentation of organs and tumors from abdominal CT images.  Building upon nnU-Net, the proposed approach addresses abdominal organ and whole-body pan-tumor segmentation for both abdominal and whole-body CT images. The method consists of multiple stages.

The writing and organization of this paper are commendable. The authors have effectively presented their research in a clear and concise manner, making it easy for readers to follow and understand the proposed method.

questions:

1. The workflow of the proposed approach in figure 1  looks complicated. It should be improved for a better understanding.

---

> ### Author Response · Authors · 2023-11-06
> **Update Figure 1**
>
> We have updated Figure 1 for a better understanding.

---

> > ### Comment · Reviewer_pJUa · 2023-12-01
> >
> > The authors have satisfactorily addressed my concern.

---

### Decision · Program_Chairs · 2023-10-24

Accept